# Rat sensitivity to multipoint statistics is predicted by efficient coding of natural scenes

**Riccardo Caramellino[1†], Eugenio Piasini[2†‡], Andrea Buccellato[1], Anna Carboncino[1], Vijay Balasubramanian[2]\*, Davide Zoccolan[1]\***

[1]Visual Neuroscience Lab, International School for Advanced Studies, Trieste, Italy; [2]Computational Neuroscience Initiative, University of Pennsylvania, Philadelphia, United States

**\*For correspondence:**
vijay@physics.upenn.edu (VB);
zoccolan@sissa.it (DZ)

†These authors contributed equally to this work

**Present address:** ‡Neural Computation Lab, International School for Advanced Studies, Trieste, Italy

**Competing interest:** The authors declare that no competing interests exist.

**Abstract** Efficient processing of sensory data requires adapting the neuronal encoding strategy to the statistics of natural stimuli. Previously, in Hermundstad et al., 2014, we showed that local multipoint correlation patterns that are most variable in natural images are also the most perceptually salient for human observers, in a way that is compatible with the efficient coding principle. Understanding the neuronal mechanisms underlying such adaptation to image statistics will require performing invasive experiments that are impossible in humans. Therefore, it is important to understand whether a similar phenomenon can be detected in animal species that allow for powerful experimental manipulations, such as rodents. Here we selected four image statistics (from single- to four-point correlations) and trained four groups of rats to discriminate between white noise patterns and binary textures containing variable intensity levels of one of such statistics. We interpreted the resulting psychometric data with an ideal observer model, finding a sharp decrease in sensitivity from two- to four-point correlations and a further decrease from four- to three-point. This ranking fully reproduces the trend we previously observed in humans, thus extending a direct demonstration of efficient coding to a species where neuronal and developmental processes can be interrogated and causally manipulated.

## Editor's evaluation

This work will be of interest to neuroscientists who want to understand how visual systems are tuned to and encode natural scenes. It reports that rats share phenomenology with humans in sensitivity to spatial correlations in scenes. This shows that an earlier paper's hypothesis about efficient coding may be more broadly applicable. This work also opens up the possibility of studying this kind of visual tuning in an animal where invasive techniques can be used to study this neural origins of this sensitivity and its development.

## Introduction

It is widely believed that the tuning of sensory neurons is adapted to the statistical structure of the signals they must encode (*Sterling and Laughlin, 2015*). This normative principle, known as efficient coding, has been successful in explaining many aspects of neural processing in vision (*Atick and Redlich, 1990*; *Fairhall et al., 2001*; *Laughlin, 1981*; *Olshausen and Field, 1996*; *Pitkow and Meister, 2012*), audition (*Carlson et al., 2012*; *Smith and Lewicki, 2006*) and olfaction (*Teşileanu et al., 2019*), including adaptation (*Mlynarski and Hermundstad, 2021*) and gain control (*Schwartz and Simoncelli, 2001*). In *Hermundstad et al., 2014*, we reported that human sensitivity to visual

**Figure 1.** Visual stimuli and behavioral task. (**A**) Schematic of the four kinds of texture discrimination tasks administered to the four groups of rats in our study. Each group had to discriminate unstructured binary textures containing white noise (example on the left) from structured binary textures containing specific types of local multipoint correlations among nearby pixels (i.e. 1-, 2-, 3-, or 4-point correlations; examples on the right). The textures were constructed to be as random as possible (maximum entropy), under the constraint that the strength of a given type of correlation matched a desired level. The strength of a correlation pattern was quantified by the value (intensity) of a corresponding statistic (see main text), which could range from 0 (white noise) to 1 (maximum possible amount of correlation). The examples shown here correspond to intensities of 0.85 (one- and two-point statistics) and 0.95 (three- and four-point statistics). (**B**) Schematic representation of a behavioral trial. Left and center: animals initiated the presentation of a stimulus by licking the central response port placed in front of them. This prompted the presentation of either a structured (top) or an unstructured (bottom) texture. Right: in order to receive the reward, animals had to lick either the left or right response port to report whether the stimulus contained the statistic (top) or the noise (bottom). *Figure 1—figure supplement 1* shows the performances attained by four example rats (one per group) during the initial phase of the training (when the animals were required to discriminate the stimuli shown in A), as well as the progressively lower statistic intensity levels that these rats progressively learned to discriminate from white noise during the second phase of the experiment.

The online version of this article includes the following figure supplement(s) for figure 1:

**Figure supplement 1.** Learning curves of four example rats during the two initial training phases.

textures defined by local multipoint correlations depends on the variability of such correlations across natural scenes. This allocation of resources to features that are the most variable in the environment, and thus more informative about its state, is accounted for by efficient coding, demonstrating its role as an organizing principle also at the perceptual level (*Hermundstad et al., 2014*; *Tesileanu et al., 2020*; *Tkacik et al., 2010*). However, it remains unknown whether this preferential encoding of texture statistics that are the most variable across natural images is a general principle underlying visual perceptual sensitivity across species. Although some evidence exists for differential neural encoding of multipoint correlations in macaque V2 (*Yu et al., 2015*) and V1 (*Purpura et al., 1994*), the sensitivity ranking we previously reported in *Hermundstad et al., 2014* has not been investigated in any species other than humans (*Hermundstad et al., 2014*; *Tesileanu et al., 2020*; *Tkacik et al., 2010*; *Victor and Conte, 2012*). Moreover, while monkeys are standard models of advanced visual processing (*DiCarlo et al., 2012*; *Kourtzi and Connor, 2011*; *Lehky and Tanaka, 2016*; *Nassi and Callaway, 2009*; *Orban, 2008*), they are less amenable than rodents to causal manipulations (e.g. optogenetic or controlled rearing) to interrogate how neural circuits may adapt to natural image statistics. On the other hand, rodents have emerged as powerful model systems to study visual functions during the last decade (*Glickfeld et al., 2014*; *Glickfeld and Olsen, 2017*; *Huberman and Niell, 2011*; *Katzner and Weigelt, 2013*; *Niell and Scanziani, 2021*; *Reinagel, 2015*; *Zoccolan, 2015*). Rats, in particular, are able to employ complex shape processing strategies at the perceptual level (*Alemi-Neissi et al., 2013*; *De Keyser et al., 2015*; *Djurdjevic et al., 2018*; *Vermaercke and Op de Beeck, 2012*), and rat lateral extrastriate cortex shares many defining features with the primate ventral stream (*Kaliukhovich and Op de Beeck, 2018*; *Matteucci et al., 2019*; *Piasini et al., 2021*; *Tafazoli et al., 2017*; *Vermaercke et al., 2014*; *Vinken et al., 2017*). More importantly, it was recently shown that rearing newborn rats in controlled visual environments allows causally testing long-standing hypotheses about the dependence of visual cortical development from natural scene statistics (*Matteucci and Zoccolan,*

*2020*). Establishing the existence of a preferential encoding of less predictable statistics in rodents is therefore crucial to understand the neural substrates of efficient coding and its relationship with postnatal visual experience.

## Results

To address this question, we measured rat sensitivity to visual textures defined by local multipoint correlations, training the animals to discriminate binary textures containing structured noise from textures made of white noise (*Figure 1A*). The latter were generated by independently setting each pixel to black or white with equal probability, resulting in no spatial correlations. Structured textures, on the other hand, were designed to enable precise control over the type and intensity of the correlations they contained. To generate these textures we built and published a software library (*Piasini, 2021*) that implements the method developed in *Victor and Conte, 2012*. Briefly, for any given type of multipoint correlation (also termed a statistic in what follows), we sampled from the distribution over binary textures that had the desired probability of occurrence of that statistic, but otherwise contained the least amount of structure (i.e. had maximum entropy). The probability of occurrence of the pattern was parametrized by the intensity of the corresponding statistic, determined by a parity count of white or black pixels inside tiles of 1, 2, 3, or 4 pixels (termed gliders) used as the building blocks of the texture (*Victor and Conte, 2012*). When the intensity is zero, the texture does not contain any structure–it is the same as white noise (*Figure 1A*, left). When the intensity is +1, every possible placement of the glider across the texture contains an even number of white pixels, while a level of –1 corresponds to all placements containing an odd number of white pixels. Intermediate intensity levels correspond to intermediate fractions of gliders containing the even parity count. The structure of the glider and the sign of the intensity level dictate the appearance of the final texture. For instance (see examples in *Figure 1A*, right), for positive intensity levels, a one-point glider produces textures with increasingly large luminance, a two-point glider produces oriented edges and a four-point glider produces rectangular blocks. A three-point glider produces L-shape patterns, either black or white depending on whether the intensity is negative or positive.

Notably, two-point and three-point gliders are associated to multiple distinct multipoint correlations, corresponding to different spatial glider configurations. For instance, two-point correlations can arise from horizontal (-), vertical (|) or oblique gliders (/, \), while three-point correlations can give rise to L patterns with various orientations ($\theta_\neg$, $\theta_\vdash$, $\theta_\llcorner$, $\theta_\lrcorner$). In our previous study with human participants (*Hermundstad et al., 2014*), we tested all these two-point, three-point, and four-point configurations, as well as 11 of their pairwise combinations, for a total of 20 different texture statistics. In that set of experiments, we did not test textures defined by one-point correlations because, by construction, the method we used to measure the variability of texture statistics across natural images could not be applied to the one-point statistic. In our current study, practical and ethical constraints prevented us from measuring rat sensitivity to a large number of statistic combinations, because a different group of animals had to be trained with each tested statistic (see below), meaning that the number of rats required for the experiments increased rapidly with the number of statistics studied. Therefore, we chose to test the 4-point statistic, as well as one each of the two-point and three-point statistics (those shown in *Figure 1A*). One of the three-point statistics (corresponding to the glider $\theta_\lrcorner$) was randomly selected among the four available, since in our previous study no difference was found among the variability of distinct three-point textures across natural images, and aggregate human sensitivity to three-point correlations was measured without distinguishing among glider configurations. As for the two-point statistic, we selected one of the two gliders (the horizontal one) that yielded the largest sensitivity in humans, so as to include in our stimulus set at least an instance of both the most discriminable (two-point -) and least discriminable (three-point $\theta_\lrcorner$) textures. In addition, we also tested the one-point statistic because, given the well-established sensitivity of the rat visual system to luminance changes (*Minini and Jeffery, 2006*; *Tafazoli et al., 2017*; *Vascon et al., 2019*; *Vermaercke and Op de Beeck, 2012*), performance with this statistic served as a useful benchmark against which to compare rat discrimination of the other, more complex textures. Finally, while in *Hermundstad et al., 2014*, both positive and negative values of the statistics were probed against white noise, here we tested only one side of the texture intensity axis (either positive, for one-, two-, and four-point configurations, or negative, for three-point ones) — again, with the goal of limiting the number of rats used in the experiment (see Materials and methods for more details on the rationale behind the choice

of statistics and their polarity, and see Discussion for an assessment of the possible impact of these choices on our conclusions).

For each of the four selected image statistics, we trained a group of rats to discriminate between white noise and structured textures containing that statistic with nonzero intensity (*Figure 1A*). Each trial of the experiment started with the rat autonomously triggering the presentation of a stimulus by licking the central response port within an array of three (*Figure 1B*). The animal then reported whether the texture displayed over the monitor placed in front of him contained the statistic (by licking the left port) or white noise (by licking the right port). The rat received liquid reward for correct choices and was subjected to a time-out period for incorrect ones (*Figure 1B*). In the initial phase of the experiment, the intensity of the statistic was set to a single level, close to the maximum (or minimum, in case of the three-point statistic, for which we used only negative values), to make the discrimination between structured textures and white noise as easy as possible for naive rats that had to learn the task from scratch. The learning curves of four example rats, one per group, are shown in *Figure 1—figure supplement 1A*. In the following phase of the experiment, the intensity of the statistic was gradually reduced using an adaptive staircase procedure (see Materials and methods) to make the task progressively harder. The asymptotic levels of the statistics reached across consecutive training sessions by four example rats, one per group, are shown in *Figure 1—figure supplement 1B*. Following this training, rats were subjected to: (1) a main testing phase, where textures were sampled at regular intervals along the intensity level axis and were randomly presented to the animals; and (2) a further testing phase, where rats originally trained with a given statistic were probed with a different one (see Materials and methods for details on training and testing).

The main test phase yielded psychometric curves showing the sensitivity of each animal in discriminating white noise from the structured texture with the assigned statistic (example in *Figure 2A*, black dots). To interpret results, we developed an ideal observer model, in which the presentation of a texture with a level of the statistic equal to s produces a percept $x$ sampled from a truncated Gaussian distribution centered on the actual value of the statistic ($s$) with a fixed standard deviation σ (*Fleming et al., 2013*; *Geisler, 2011*). Here, σ measures the 'blurriness' in the animal's sensory representation for a particular type of statistic (i.e. the perceptual noise) and, consequently, its inverse 1/σ captures its resolution, or sensitivity — i.e., the perceptual threshold for discriminating a structured texture from white noise. As detailed in the Materials and methods, our ideal observer model yields the psychometric function giving the probability of responding 'noise' at any given level of the statistic $s$ as

$$p(\text{report noise}|s) = \frac{\Phi\left(\frac{x^*(\alpha,\sigma)-s}{\sigma}\right) - \Phi\left(\frac{-1-s}{\sigma}\right)}{\Phi\left(\frac{1-s}{\sigma}\right) - \Phi\left(\frac{-1-s}{\sigma}\right)}$$

where $\Phi(x)$ is the standard Normal cumulative density function, $\alpha$ captures the animal's prior choice bias and $x^*(\alpha, \sigma)$ is the decision boundary used by the animal to divide the perceptual axis into 'noise' and 'structured texture' regions. The two free parameters of the model (α and σ) parameterize the psychometric function (example in *Figure 2A*, blue curve) and can be estimated from behavioral data by maximum likelihood. Prior bias ($\alpha$) and sensitivity ($1/\sigma$) are related, respectively, to the horizontal offset and slope of the curve.

Fitting this model to the behavioral choices of rats in the four groups led to psychometric functions with a characteristic shape, which depended on the order of the multipoint statistic an animal had to discriminate (*Figure 2B*). In particular, the sensitivity 1/σ followed a specific ranking among the groups (*Figure 2C*), being higher for one- and two-point than for three-point ($p_1<0.001$ and $p_2<0.001$, two-sample t-test with Holm-Bonferroni correction) and four-point ($p_1<0.001$, $p_2<0.001$) correlations, and larger for four-point than three-point correlations ($p<0.01$). When focusing on the texture statistics that had been also tested in our previous study (i.e. two-point horizontal, three-point, and four-point correlations), this sensitivity ranking was the same as the one observed in humans and as the variability ranking measured across natural images (*Hermundstad et al., 2014*): two-point horizontal > four-point > three-point. Moreover, for the set of statistics that were studied both here and in *Hermundstad et al., 2014*, the actual values of the rat sensitivity matched, up to a scaling factor, both the human sensitivity and the standard deviation of the statistics in natural images (*Figure 3*). This match was quantified with the 'degree of correspondence', defined in *Hermundstad et al., 2014*, which takes on values between 0 and 1, with one indicating perfect quantitative match (see Materials and methods for details). The degree of correspondence was 0.986 between rat sensitivity and image

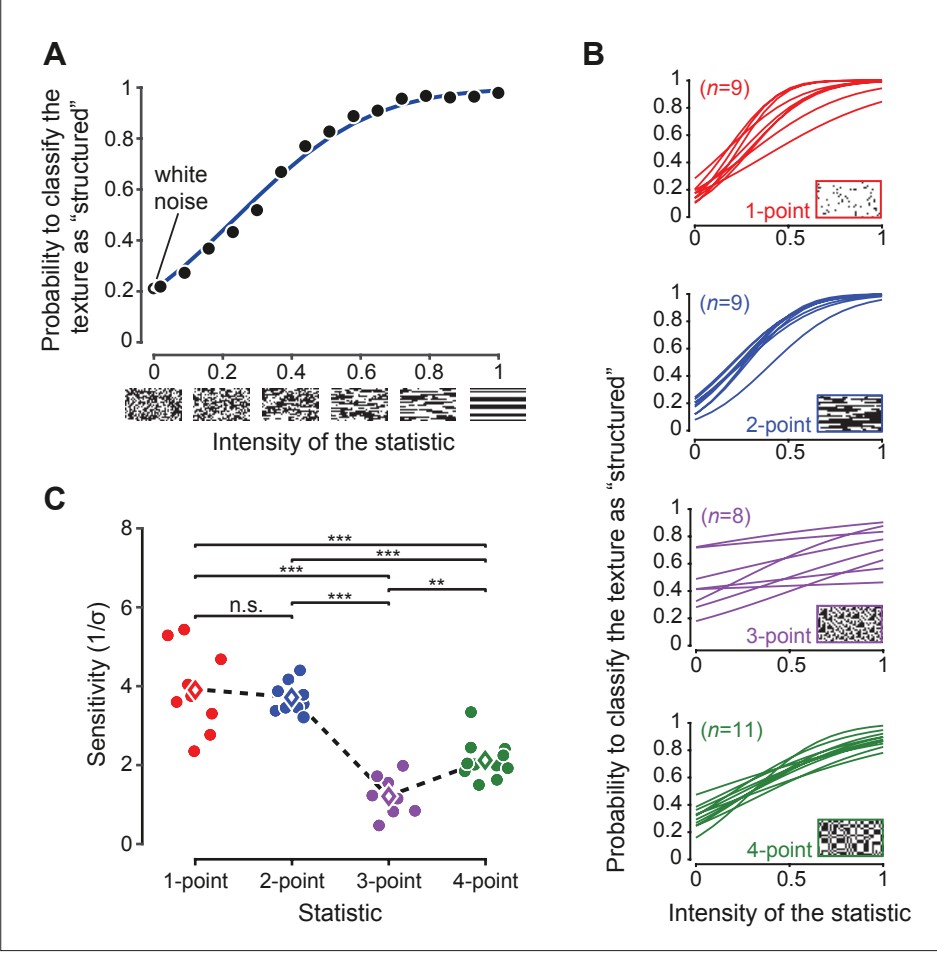

**Figure 2.** Rat sensitivity to multipoint correlations. (**A**) Psychometric data for an example rat trained on two-point correlations. Black dots: fraction of trials in which a texture with the corresponding intensity of the statistic was correctly classified as 'structured'. Empty black circle: fraction of trials the rat has judged a white noise texture as containing the statistic. Blue line: psychometric function corresponding to the fitted ideal observer model (see main text). (**B**) Psychometric functions obtained for all the rats tested on the four statistics (n indicates the number of animals in each group). (**C**) Values of the perceptual sensitivity $1/\sigma$ to each of the four statistics. Filled dots: individual rat estimates. Empty diamonds: group averages. The dashed line emphasizes the sensitivity ranking observed for the four statistics. Significance markers ** and *** indicate, respectively, p < 0.01 and p < 0.001 for a two-sample t-test with Holm-Bonferroni correction. The same analysis was repeated in *Figure 2—figure supplement 1* including only the rats that reached a certain performance criterion during the initial training.

The online version of this article includes the following figure supplement(s) for figure 2:

**Figure supplement 1.** Rat sensitivity to multipoint correlations after excluding animals that did not reach the 65% correct discrimination criterion in phase I of the training.

statistics (p-value: 0.07, Monte Carlo test), and 0.990 between rat sensitivity and human sensitivity (p-value: 0.05, Monte Carlo test). For reference, *Hermundstad et al., 2014* reported values between 0.987 and 0.999 for the degree of correspondence between human sensitivity and image statistics. This indicates not only a qualitative but also a quantitative agreement between our findings and the pattern of texture sensitivity predicted by efficient coding.

To further validate these findings, we performed additional within-group and within-subject comparisons. To this end, each group of animals was either tested with a new statistic or was split into two subgroups, each tested with a different statistic. Results of these additional experiments are reported in *Figure 4*, comparing the sensitivity to the new statistic(s) with the sensitivity to the originally learned statistic (colored symbols without and with halo, respectively) for each group/subgroup. Rats trained on one- and two-point statistics (the most discriminable ones; see *Figure 2C*) performed

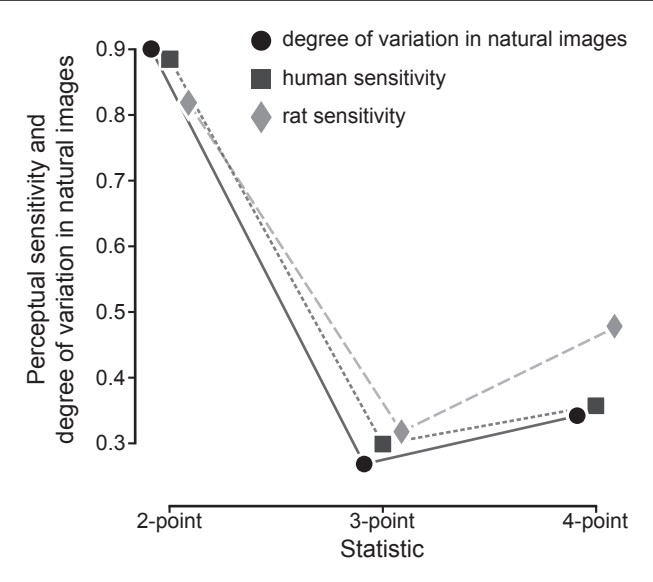

**Figure 3.** Quantitative match between rat sensitivities, human sensitivities and texture variability across natural images. For the three texture statistics that were tested both in our current study with rats and in our earlier study with humans (***Hermundstad et al., 2014***), rat average sensitivities are compared to human average sensitivities and to the variability of these statistics across natural scenes data from ***Hermundstad et al., 2014***. The three sets of data points have been scaled in such a way that each triplet of sensitivity values had Euclidean norm = 1, so as to allow an easier qualitative comparison (for a quantitative comparison see main text).

poorly with higher-order correlations (compare the green and purple star with the red star, and the green and purple cross with the blue cross in ***Figure 4***), while animals trained on the four-point statistic performed on two-point correlations as well as rats that were originally trained on those textures (compare the blue square to the blue cross). This shows that the better discriminability of textures containing lower order correlations is a robust phenomenon, which is independent of the history of training and observable within individual subjects. Moreover, performance on four-point correlations was higher than performance on three-point correlations for each group of rats (compare the green to the purple symbols connected by a line). This was true, in particular, not only for rats trained on four-point and switching to three-point (green vs. purple square, p < 0.01, paired one-tailed t-test) but even for rats trained on three-points and switching to four-point (green vs. purple triangle, p < 0.05, paired one-tailed t-test). This means that the larger discriminability of the four-point statistic, as compared to the three-point one, is a statistically robust phenomenon within individual subjects.

## Discussion

Overall, our results show that rat sensitivity to multipoint statistics is similar to the one we previously observed in humans and to the variability of multipoint correlations we previously measured across natural images (***Hermundstad et al., 2014***; ***Tkacik et al., 2010***). This agreement holds both qualitatively and quantitatively (***Figures 2–4***). Importantly, we found the expected sensitivity ranking (two-point horizontal > four-point > three-point) to be robust not only across groups (***Figure 2C***) but also for animals that were sequentially tested with multiple texture statistics (***Figure 4***) - and even at the within-subject level for the crucial three-point vs. four-point comparison. Moreover, we found a high degree of correspondence between rat and human sensitivities (***Figure 3***).

A potential limitation of our study is related to our stimulus choices, both in terms of selected texture statistics and polarity (i.e. negative vs. positive intensity). A first possible issue is whether the three texture statistics that were tested in both the present study and in ***Hermundstad et al., 2014*** are sufficient to allow a meaningful comparison between rat and human sensitivities, as well as rat sensitivity and texture variability in natural scenes. We addressed this matter at the level of experimental design, by carefully choosing the three statistics that, based on the sensitivity ranking

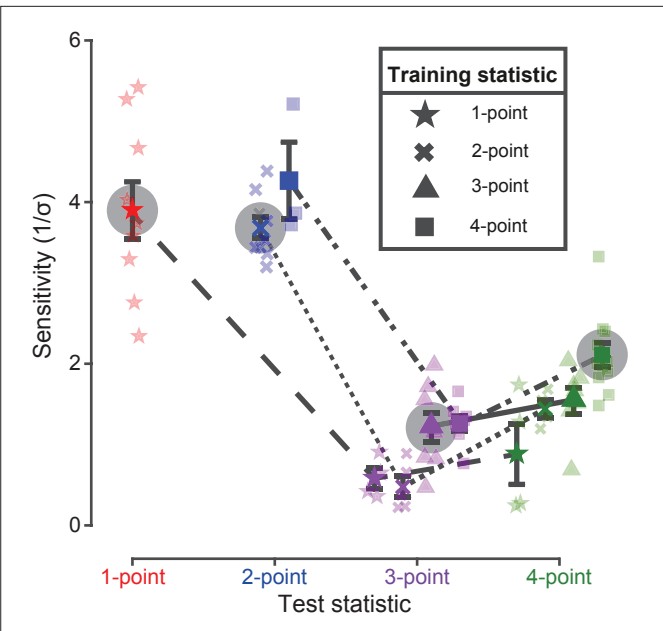

**Figure 4.** Rat sensitivity to multipoint correlations – dependence on training history and within-subject analysis. Colored points with halo show the average sensitivities (with SEM) of the four groups of rats to the statistics (indicated by the symbols in the key) they were originally trained on (i.e. same data as the colored diamonds in *Figure 2C*). The other colored symbols connected by a line show the average sensitivities (with SEM) obtained when subgroups of rats originally trained on a given statistic (as indicated by the symbol in the key) were tested with different statistics (as indicated in abscissa). Specifically: (1) out of the nine rats originally trained/tested with one-point correlations (star), four were tested with three-point (purple star) and four with four-point (green star) correlations (one rat did not reach this test phase); (2) out of the nine rats originally trained/tested with two-point correlations (cross), five were tested with three-point (purple cross) and four with four-point (green cross) correlations; (3) out of the eight rats originally trained/tested with three-point correlations (triangle), seven were tested with four-point (green triangle) correlations (one rat did not reach this test phase); and (4) out of the eleven rats originally trained/tested with four-point correlations (square), eight were tested with three-point (purple square) and three with two-point correlations (blue square). Sensitivities achieved by individual animals are represented as shaded data points with the corresponding symbol/color combination.

observed in humans, would have yielded the cleanest signature of efficient coding (*Hermundstad et al., 2014*). That is, we selected two statistics that were, respectively, maximally and minimally variable across natural images, and yielded the largest and lowest sensitivities in humans: horizontal two-point correlations and one of the three-point correlations. The four-point correlation was then a natural choice as the third statistic, as it was the only one characterized by a differently shaped glider. Additionally, human sensitivity to this statistic, as well as its variability across natural images, is only slightly larger than for the three-point configurations. Therefore, finding a reliable sensitivity difference between three-point and four-point textures also for rats would have provided strong evidence for matching texture sensitivity across the two species. Due to the experimental limitations discussed in the Results and the Materials and methods sections, we were unable to analyze one of the oblique two-point statistics, for which human sensitivity takes on an intermediate value between the two-point horizontal and three-point correlations, and that in humans allows one to differentiate between the predictions of efficient coding and those stemming from an oblique effect for patterns that are rotated versions of each other (*Hermundstad et al., 2014*).

The second potential limitation is related to the choice of polarity (positive or negative intensity values for the examined statistics). This choice was guided by different considerations depending on the kind of statistic. For one-point correlations we chose positive intensity values because they yield patterns that are brighter than white noise. Since previous work from our group has shown that rat V1 neurons are very sensitive to increases of luminance (*Tafazoli et al., 2017*; *Vascon et al., 2019*), our choice ensured that one-point textures were highly distinguishable from white noise (as indeed observed in our data; see *Figure 2B–C*), which was the key requirement for our benchmark statistic.

This enabled us to guard against issues in our task design: if the animals had failed to discriminate one-point textures, this would have suggested an overall inadequacy of the behavioral task rather than a lack of perceptual sensitivity to luminance changes. For two-point and four-point statistics we also used positive intensity values — a choice dictated by the need of testing a rodent species that has much lower visual acuity than humans (*Keller et al., 2000*; *Prusky et al., 2002*; *Zoccolan, 2015*). Positive two-point and four-point correlations give rise to large features (thick oriented stripes and wide rectangular blocks made of multiple pixels with the same color), while negative intensities produce higher spatial frequency patterns, where color may change every other pixel (see Figure 2A in *Hermundstad et al., 2014*). Therefore, using negative two-point and four-point statistics would have introduced a possible confound, since low sensitivity to these textures could have been simply due to the low spatial resolution of rat vision. For three-point correlations, polarity does not affect the shape and size of the emerging visual patterns, but it determines their contrast. Positive and negative intensities yield L-shaped patches that are, respectively, white and black. In this case, we chose the latter to make sure that the well-known dominance of OFF responses observed across the visual systems of many mammal species would not play in favor of finding the lowest sensitivity for the three-point statistic. In fact, several studies have shown that primary visual neurons of primates and cats respond more strongly to black than to white spots and oriented bars (*Liu and Yao, 2014*; *Xing et al., 2010*; *Yeh et al., 2009*). A very recent study has shown that this is the case also for the central visual field of mice, although in the periphery OFF and ON response are more balanced (*Williams et al., 2021*). Indeed, the asymmetry begins already in the retina where there are more OFF cells than ON cells (*Ratliff et al., 2010*). Since in our behavioral rigs rats face frontally the stimulus display (*Figure 1B*) and maintain their head oriented frontally during stimulus presentation (*Vanzella et al., 2019*), it was important that the L-shaped patterns produced by three-point correlations had the highest saliency. Choosing negative intensity values ensured that this was the case, thus excluding the possibility that the low-sensitivity found for three-point textures (*Figures 2–4*) was partially due to presentation at a suboptimal contrast. Notwithstanding these considerations, one could wonder whether probing also the opposite polarities of those tested in our study would be desirable for a tighter test of the efficient coding principle. Previous studies, however, found human sensitivity to be nearly identical for negative and positive intensity variations of each of the statistic tested in our study: one-point, two-point, three-point, and four-point correlations (*Victor and Conte, 2012*), even in the face of asymmetries of the distribution of the corresponding statistic in natural images (see Figure 3—figure supplement 9 in *Hermundstad et al., 2014*). In the present work, we have accordingly decided to focus the available resources on the differences between different statistics, rather than between positive and negative intensities of the same statistic.

In summary, our choices of texture types and their polarity were all dictated by the need of adapting to a rodent species texture stimuli that, so far, have only been used in psychophysics studies with humans (*Hermundstad et al., 2014*; *Tesileanu et al., 2020*; *Tkacik et al., 2010*; *Victor and Conte, 2012*) and neurophysiology studies in monkeys (*Purpura et al., 1994*; *Yu et al., 2015*). Our goal was to maximize the sensitivity of the comparison with humans and natural image statistics, while reducing the possible impact of phenomena (such as rat low visual acuity and the dominance of OFF responses) that could have acted as confounding factors. Thanks to these measures, our findings provide a robust demonstration that a rodent species and humans are similarly adapted to process the statistical structure of visual textures, in a way that is consistent with the computational principle of efficient coding. This attests to the fundamental role of natural image statistics in shaping visual processing across species, and opens a path toward a causal test of efficient coding through the altered-rearing experiments that small mammals, such as rodents, allow (*Hunt et al., 2013*; *Matteucci and Zoccolan, 2020*; *White and Fitzpatrick, 2007*).

## Materials and methods
### Psychophysics experiments
#### Subjects
A total of 42 male adult Long Evans rats (Charles River Laboratories) were tested in a visual texture discrimination task. Animals started the training at 10 weeks, after 1 week of quarantine upon arrival in our institute and 2 weeks of handling to familiarize them with the experimenters. Their weight at

arrival was approximately 300 g and they grew to over 600 g over the time span of the experiment. Rats always had free access to food but their access to water was restricted in the days of the behavioral training (5 days a week). They received 10–20 ml of diluted pear juice (1:4) during the execution of the discrimination task, after which they were also given free access to water for the time needed to reach at least the recommended 50 ml/kg intake per day.

The number of rats was chosen in such a way to yield meaningful statistical analyses (i.e. to have about 10 subjects for each of the texture statistic tested in our study), under the capacity constraint of our behavioral rig. The rig allows to simultaneously test six rats, during the course of 1–1.5 hr (*Zoccolan, 2015*; *Djurdjevic et al., 2018*). Given the need for testing four different texture statistics, we started with a first batch of 24 animals (i.e. 6 per statistics), which required about 6 hr of training per day. This first batch was complemented with a second one of 18 more rats, again divided among the four statistics (see below for details), so as to reach the planned number of about 10 animals per texture type. The first batch arrived in November 2018 and was tested throughout most of 2019; the second group arrived in September 2019 and was tested throughout most of 2020. In the first batch, four animals did not reach the test phase (i.e. the phase yielding the data shown in *Figure 2A and B*), because three of them did not achieve the criterion performance during the initial training phase (see below) and one died shortly after the beginning of the study. In the second batch, one rat died before reaching the test phase and two more died before the last test phase with switched statistics (i.e. the phase yielding the data of *Figure 2C*).

All animal procedures were conducted in accordance with the international and institutional standards for the care and use of animals in research and were approved by the Italian Ministry of Health and after consulting with a veterinarian (Project DGSAF 25271, submitted on December 1, 2014 and approved on September 4, 2015, approval 940/2015-PR).

## Experimental setup

Rats were trained in a behavioral rig consisting of two racks, each equipped with three operant boxes (a picture of the rig and a schematic of the operant box can be found in previous studies [*Zoccolan, 2015*; *Djurdjevic et al., 2018*]). Each box was equipped with a 21.5" LCD monitor (ASUS VEZZHR) for the presentation of the visual stimuli and an array of three stainless-steel feeding needles (Cadence Science), serving as response ports. To this end, each needle was connected to a led-photodiode pair to detect when the nose of the animal approached and touched it (a Phidgets 1203 input/output device was used to collect the signals of the photodiodes). The two lateral feeding needles were also connected to computer-controlled syringe pumps (New Era Pump System NE-500) for delivery of the liquid reward. In each box, one of the walls bore a 4.5 cm-diameter viewing hole, so that a rat could extend its head outside the box, face the stimulus display (located at 30 cm from the hole) and reach the array with the response ports.

## Choice of image statistics to be used in the experiment

As mentioned in the main text, in our experiment we studied the 1-point and 4-point statistic, as well as one of the two-point and one of the three-point statistics. In the nomenclature introduced by *Victor and Conte, 2012*, these are, respectively, the $\gamma$, $\alpha$, $\beta_-$ and $\theta_\lrcorner$ statistics. By comparison, in humans, *Victor and Conte, 2012* studied a total of five statistics (the same we tested, plus $\beta_|$), while *Hermundstad et al., 2014* tested many more, including combinations of statistic pairs, although they did not investigate $\gamma$. Our choice of which statistics to test was constrained on practical and ethical grounds by the need to use the minimum possible number of animals in our experiments, which led us to study one representative statistic per order of the glider. We note also that we decided to test the $\gamma$ statistic, even though this was omitted by *Hermundstad et al., 2014* (as explained in that paper, the method used to assess the variability of all other multipoint correlation patterns in natural images can't be applied to $\gamma$ by construction, because the binarization threshold used for images is such that $\gamma = 0$ for all images in the dataset). The reason for including $\gamma$ was that it provided a useful control on the effectiveness of our experimental design, as (unlike for the other visual patterns) we expected rats to be able to easily discriminate stimuli differing by average luminosity (*Minini and Jeffery, 2006*; *Tafazoli et al., 2017*; *Vascon et al., 2019*; *Vermaercke and Op de Beeck, 2012*). As mentioned in the Discussion, failure of the rats to discriminate one-point textures would have indicated a likely issue in the design of the task.

Human sensitivity to multipoint correlation patterns does not distinguish between positive and negative values of the statistics (*Victor and Conte, 2012*). Therefore, again in order to minimize the number of animals necessary to the experiment, we only collected data for positive values of the $\gamma$, $\beta$ and $\alpha$ statistics, and negative values of the $\theta_\lrcorner$ statistic (see below for the specific values used). Unlike two- or four-point statistics, $\theta$ statistics change contrast under a sign change (namely, positive $\theta$ values correspond to white triangular patterns on a black background, and negative $\theta$ values correspond to black triangular patterns on a white background). On the other hand, dominance of OFF responses (elicited by dark spots on a light background) has been reported in mammals, including primates, cats, and rodents (*Ratliff et al., 2010*; *Liu and Yao, 2014*; *Xing et al., 2010*; *Yeh et al., 2009*; *Williams et al., 2021*). Therefore we reasoned that if rats, unlike humans, were to have a different sensitivity to positive and negative $\theta$ values, the sensitivity to negative $\theta$ would be the higher of the two.

Finally, for the sake of simplicity, whenever in the text we refer to the 'intensity' of a statistic, this should be interpreted as the absolute value of the intensity as defined by *Victor and Conte, 2012*. This has no effect when describing $\gamma$, $\beta$, or $\alpha$ statistics, and only means that any value reported for $\theta_\lrcorner$ should be taken with a sign flip (i.e. negative instead of positive values) if trying to connect formally to the system of coordinates in *Victor and Conte, 2012*.

## Visual stimuli

Maximum-entropy textures were generated using the methods described by *Victor and Conte, 2012*. To this end, we implemented a standalone library and software package that we have since made publicly available as free software (*Piasini, 2021*). In the experiment, we used white noise textures as well as textures with positive levels of four different multipoint statistics, as described above (see also *Figure 1A*). It should be noted that, with the exception of the extreme value of the $\gamma$ statistic ($\gamma = 1$ corresponds to a fully white image), the intensity level of a given statistic does not specify deterministically the resulting texture image. In our experiment, for any intensity level of each statistic, multiple, random instances of the textures were built to be presented to the rats during the discrimination task (see below for more details).

Subjects had to discriminate between visual textures containing one of the four selected statistics and white noise. Each texture had a size of 39 × 22 pixels and occupied the entire monitor (full-field stimuli). The pixels had a dimension of about 2 degrees of visual angle. Given that the maximal resolution of rat vision is about one cycle per degree (*Keller et al., 2000*; *Prusky et al., 2000*; *Prusky et al., 2002*), such a choice of the pixel size guaranteed that the animals could discriminate between neighboring pixels of different color. Textures were showed at full-contrast over the LCD monitors that were calibrated in such a way to have minimal luminance of 0.126 ± 0.004 cd/mm (average ± SD across the six monitors), maximal luminance of 129 ± 5 cd/mm, and an approximately linear luminance response curve.

## Discrimination task

Each rat was trained to: (1) touch the central response port to trigger stimulus presentation and initiate a behavioral trial; and (2) touch one of the lateral response ports to report the identity of the visual stimulus and collect the reward (all the animals were trained with the following stimulus/response association: structured texture → left response port; white noise texture → right response port). The stimulus remained on the display until the animal responded or for a maximum of 5 s, after which the trial was considered as ignored. In case of a correct response the stimulus was removed, a positive reinforcement sound was played and a white (first animal batch) or gray (second batch) background was shown during delivery of the reward. In case of an incorrect choice, the stimulus was removed and a 1–3 s time-out period started, during which the screen flickered from middle-gray to black at a rate of 10 Hz, while a 'failure' sound was played. During this period the rat was not allowed to initiate a new trial. To prevent the rats from making impulsive random choices, trials where the animals responded in less than 300 or 400 ms were considered as aborted: the stimulus was immediately removed and a brief sound was played. In each trial, the visual stimuli had the same probability (50%) of being sampled from the pool of white noise textures or from the pool of structured textures, with the constraint that stimuli belonging to the same category were shown for at most $n$ consecutive trials (with $n$ varying between 2 and 3 depending on the animal and on the session), so as to prevent the rats from developing a bias toward one of the response ports.

Stimulus presentation, response collection and reward delivery were controlled via workstations running the open source suite MWorks (https://mworks.github.io;*Starwarz and Cox, 2021*).

## Experimental design

Each rat was assigned to a specific statistic, from one- to four-point, for which it was trained in phases I and II and then tested in phase III. Generalization to a different statistic from the one the rat was trained on was assessed in phase IV. Out of the 42 rats, 9 were trained with one-point statistics, 9 with two-point, 12 with three-point, and 12 with four-point. The animals that reached phase III were 9, 9, 8, and 11, respectively, for the four statistics.

## Phase I

Initially, rats were trained to discriminate unstructured textures made of white noise from structured textures containing a single high-intensity level of one of the statistics (for one-point and two-point: 0.85; for three-point and four-point: 0.95). To make sure that the animals learned a general distinction between structured and unstructured textures (and not between specific instances of the two stimulus categories), in each trial both kinds of stimuli were randomly sampled (without replacement) from a pool of 350 different textures. Since the rats typically performed between 200 and 300 trials in a training session, every single texture was not shown more than once. A different pool of textures was used in each of the five days within a week of training. The same five texture pools were then used again (in the same order) the following week. Therefore, at least 7 days had to pass before a given texture stimulus was presented again to a rat.

For the first batch of rats, we moved to the second phase of the experiment all the animals that were able to reach at least an average performance of 65% correct choices over a set of 500 trials, collected across a variable number of consecutive sessions (the learning curves of four example rats from this batch, one per group, are shown in *Figure 1—figure supplement 1A*). Based on this criterion, two rats tested with three-point textures and one rat tested with four-point textures were excluded from further testing. For the second batch of rats, we decided to admit all the animals to the following experimental phases after a prolonged period of training in the first phase. In fact, we reasoned that, in case some texture statistic was particularly hard to discriminate, imposing a criterion performance in the first phase of the experiment would bias the pool of rats tested with such very difficult statistic toward including only exceptionally proficient animals. This in turn, could lead to an overestimation of rat typical sensitivity to such difficult statistic. On the other hand, the failure of a rat to reach a given criterion performance could be due to intrinsic limitations of its visual apparatus (such as a malfunctioning retina or particularly low acuity). Therefore, to make sure that our result did not depend on including in our analysis some animals of the second batch that did not reach 65% correct discrimination in the first training phase, the perceptual sensitivities were re-estimated after excluding those rats (i.e. after excluding one rat from the two-point, three rats from the three-point, and one from the four-point groups). As shown in *Figure 2—figure supplement 1*, the resulting sensitivity ranking was unchanged (compare to *Figure 2C*) and all pairwise comparisons remained statistically significant (two-sample t-test with Holm-Bonferroni correction).

## Phase II

In this phase, we introduced progressively lower levels of intensity of each statistic, bringing them gradually closer to the zero-intensity level corresponding to white noise. To this end, we applied an adaptive staircase procedure to update the minimum level of the statistic to be presented to a rat based on its current performance. Briefly, in any given trial, the level of the multipoint correlation in the structured textures was randomly sampled between a minimum level (under the control of the staircase procedure) and a maximum level (fixed at the value used in phase I). Within this range, the sampling was not uniform, but was carried out using a geometric distribution (with the peak at the minimum level), so as to make much more likely for rats to be presented with intensity levels at or close to the minimum. The performance achieved by the rats on the current minimum intensity level was computed every ten trials. If such a performance was higher than 70% correct, the minimum intensity level was decreased by a step of 0.05. By contrast, if the performance was lower than 50%, the minimum intensity level was increased of the same amount.

This procedure allowed the rats to learn to discriminate progressively lower levels of the statistic in a gradual and controlled way (the asymptotic levels of the statistics reached across consecutive training sessions by four example rats of the first batch, one per group, are shown in *Figure 1—figure supplement 1B*). At the end of this phase, the minimum intensity level reached by the animal in the three groups was: 0.21 ± 0.12, 0.2 ± 0.2, 0.70 ± 0.22, and 0.56 ± 0.18 (group average ± SD) for, respectively, one-, two-, three-, and four-point correlations.

### Phase III

After the training received in phases I and II, the rats were finally moved to the main test phase, where we measured their sensitivity to the multipoint correlations they were trained on. In each trial of this phase, the stimulus was either white noise or a patterned texture with equal probability. If it was a patterned texture, the level of the statistic was randomly selected from the set {0.02, 0.09, 0.16, …, 0.93, 1} (i.e. from 0.02 to 1 in steps of 0.07) with uniform probability. The responses of each rat over this range of intensity levels yielded psychometric curves (see example in *Figure 1B*), from which rat sensitivity was measured by fitting the Bayesian ideal observer model described below (*Figure 2A and B*).

### Phase IV

To verify the sensitivity ranking observed in phase III, we carried out an additional test phase, where each rat was tested on a new statistic, which was different from the one the animal was previously trained and tested on. The two groups of rats that were originally trained with the statistics yielding the highest sensitivity in phase III (i.e. one- and two-point correlations; see *Figure 2B*) were split in approximately equally-sized subgroups and each of these subgroups was tested with the less discriminable statistics (i.e. three- and four-point correlations; leftmost half of *Figure 2C*). This allowed assessing that, regardless of the training history, sensitivity to four-point correlations was slightly but consistently higher than sensitivity to three-point correlations. For the group of rats originally tested with the three-point statistic, all the animals were switched to the four-point (third set of points in *Figure 2C*). This allowed comparing the sensitivities to these statistics at the within-subject level (notably, these rats were found to be significantly more sensitive to the four-point textures than to the three-point, despite the extensive training they had received with the latter). For the same reason, most of the rats (8/11) of the last group (i.e. the animals originally trained/tested with the four-point correlations; last set of points in *Figure 2C*) were switched to the three-point statistic, which yielded again the lowest discriminability. A few animals (3/11) were instead tested with the two-point statistic, thus verifying that the latter was much more discriminable than the four-point one (again, despite the extensive training the animals of this group had received with the four-point textures).

### Data Availability

Experimental data are available at *Caramellino et al., 2021*.

## Ideal observer model

In this section we describe the ideal observer model we used to estimate the sensitivity of the rats to the different textures. The approach is a standard one and is inspired by that in *Fleming et al., 2013*. Because our intention is to use an ideal observer as a model for animal behavior, we will write interchangeably 'rat', 'animal', and 'ideal observer' in the following.

### Preliminaries

The texture discrimination task is a two-alternative forced choice (2AFC) task, where the stimulus can be either a sample of white noise or a sample of textured noise, and the goal of the animal is to correctly report the identity of each stimulus. On any given trial, either stimulus class can happen with equal probability. The texture class is composed of $K$ discrete, positive values of the texture. In practice, $K = 14$, and these values are $\{0.02, 0.09, \ldots, 0.93, 1\}$, but we'll use a generic $K$ in the derivations for clarity. The texture statistics are parametrised such that a statistic value of zero corresponds to white noise. Therefore, if we call $s$ the true level of the statistic, the task is a parametric discrimination task where the animal has to distinguish $s = 0$ from $s > 0$.

## Key assumptions

1. each trial is independent from those preceding and following it (both for the generated texture and for the animal's behavior);

2. on any given trial, the nominal (true) value of the statistic is some value $s$. Because the texture has finite size, the empirical value of the statistic in the texture will be somewhat different from $s$. We lump this uncertainty together with that induced by the animal's perceptual process, and we say that any given trial results on the production of a *percept* $x$, sampled from a truncated Normal distribution centered around the nominal value of the statistic and bounded between

   $$a = -1 \text{ and } b = 1 : p(x|s, \sigma, a, b) = \frac{1}{\sigma} \frac{\phi\left(\frac{x-s}{\sigma}\right)}{\Phi\left(\frac{b-s}{\sigma}\right) - \Phi\left(\frac{a-s}{\sigma}\right)}$$

   where $\phi(\cdot)$ is the probability density function of the standard Normal and $\Phi(\cdot)$ is its cumulative density function. Setting the bounds to –1 and 1 allows us to account for the fact that the value of a statistic is constrained within this range by construction. We will keep $a$ and $b$ in some of the expressions below for generality and clarity, and we will substitute their values only at the end.

3. we assume that each rat has a certain prior over the statistic level that we parametrise by the log prior odds:

   $$\alpha := \ln \frac{p(s=0)}{p(s>0)}$$

   where $\alpha$ depends on the rat. More specifically, we assume that each rat assigns a prior probability $p(s = 0) = 1/(1 + e^{-\alpha})$ to the presentation of a noise sample, and a probability of $1/[K(1 + e^{\alpha})]$ to the presentation of a texture coming from any of the $K$ nonzero statistic values. In formulae: $p(s) = \frac{\delta_{s,0}}{1+e^{-\alpha}} + \frac{1}{K} \sum_{k=1}^{K} \frac{\delta_{s,s_k}}{1+e^{\alpha}}$ where $\delta$ is Kronecker's delta, and $s_k > 0, k \in \{1, \ldots, K\}$ are the $K$ possible nonzero values of the statistic. Note that this choice of prior matches the distribution actually used in generating the data for the experiment, except that $\alpha$ is a free parameter instead of being fixed at 0.

4. we assume that the true values of $\alpha$, $\sigma$, $a$ and $b$ are accessible to the decision making process of the rat.

## Derivation of the ideal observer

For a particular percept, the ideal observer will evaluate the posterior probability of noise vs texture given that percept. It will report 'noise' if the posterior of noise is higher than the posterior of texture, and 'texture' otherwise.

More in detail, for a given percept $x$ we can define a decision variable $D$ as the log posterior ratio:

$$D(x) := \ln \frac{p(s=0|x)}{p(s>0|x)} = \ln \frac{p(x|s=0)}{p(x|s>0)} + \ln \frac{p(s=0)}{p(s>0)} \tag{1}$$

With this definition, the rat will report 'noise' when $D > 0$ and 'texture' otherwise.

By plugging in the likelihood functions and our choice of prior, we get

$$D(x) = \alpha + \ln \left[ \frac{1}{\sigma} \frac{\phi(x/\sigma)}{\Phi\left(\frac{b}{\sigma}\right) - \Phi\left(\frac{a}{\sigma}\right)} \right] - \ln \left[ \frac{1}{K} \sum_k \frac{1}{\sigma} \frac{\phi\left(\frac{x-s_k}{\sigma}\right)}{\Phi\left(\frac{b-s_k}{\sigma}\right) - \Phi\left(\frac{a-s_k}{\sigma}\right)} \right] \tag{2}$$

Now, remember that *given a value of the percept x*, the decision rule based on $D$ is fully deterministic (maximum a posteriori estimate). But on any given trial we don't know the value of the percept — we only know the nominal value of the statistic. On the other hand, our assumptions above specify the distribution $p(x|s)$ for any $s$, so the deterministic mapping $D(x)$ means that we can compute the probability of reporting 'noise' as,

$$p(\text{report noise}|s) = p(D > 0|s) = \int_{x:D(x)>0} p(x|s)x \tag{3}$$

We note at this point that $D(x)$ is monotonic: indeed,

$$\frac{D(x)}{x} = -\frac{x}{\sigma^2} +$$

$$- \left[ \frac{1}{K} \sum_k \frac{\exp\left[-\frac{(x-s_k)^2}{2\sigma^2}\right]}{\Phi\left(\frac{b-s_k}{\sigma}\right) - \Phi\left(\frac{a-s_k}{\sigma}\right)} \right]^{-1} \frac{1}{K} \sum_k \frac{\exp\left[-\frac{(x-s_k)^2}{2\sigma^2}\right]\left(-\frac{x-s_k}{\sigma^2}\right)}{\Phi\left(\frac{b-s_k}{\sigma}\right) - \Phi\left(\frac{a-s_k}{\sigma}\right)}$$

$$= -\left[ \frac{1}{K} \sum_k \frac{\exp\left[-\frac{(x-s_k)^2}{2\sigma^2}\right]}{\Phi\left(\frac{b-s_k}{\sigma}\right) - \Phi\left(\frac{a-s_k}{\sigma}\right)} \right]^{-1} \frac{1}{K} \sum_k \frac{\exp\left[-\frac{(x-s_k)^2}{2\sigma^2}\right]\frac{s_k}{\sigma^2}}{\Phi\left(\frac{b-s_k}{\sigma}\right) - \Phi\left(\frac{a-s_k}{\sigma}\right)} \tag{4}$$

$$< 0 \text{ for all } x$$

where for the last inequality we have used the fact that $a < b$ and therefore, $\Phi((b - s_k)/\sigma) > \Phi((a - s_k)/\sigma)$. This result matches the intuitive expectation that a change in percept in the positive direction (i.e. away from zero) should always make it less likely for the observer to report 'noise'.

Because $D(x)$ is monotonic, there will be a unique value of $x$ such that $D(x) = 0$, and the integration region $x : D(x) > 0$ will simply consist of all values of $x$ smaller than that. More formally, if we define

$$x^* = x^*(\alpha, \sigma) \text{ such that } D(x^*) = 0 \tag{5}$$

we can write

$$p(\text{report noise}|s) = \int_a^{x^*} \frac{1}{\sigma} \frac{\phi\left(\frac{x-s}{\sigma}\right)}{\Phi\left(\frac{b-s}{\sigma}\right) - \Phi\left(\frac{a-s}{\sigma}\right)} dx$$

$$= \frac{\Phi\left(\frac{x^*(\alpha,\sigma)-s}{\sigma}\right) - \Phi\left(\frac{a-s}{\sigma}\right)}{\Phi\left(\frac{b-s}{\sigma}\right) - \Phi\left(\frac{a-s}{\sigma}\right)} \tag{6}$$

$$= \frac{\Phi\left(\frac{x^*(\alpha,\sigma)-s}{\sigma}\right) - \Phi\left(\frac{-1-s}{\sigma}\right)}{\Phi\left(\frac{1-s}{\sigma}\right) - \Phi\left(\frac{-1-s}{\sigma}\right)}$$

where in the last passage we have substituted $a = -1$ and $b = 1$.

## Example: single-level discrimination case

To give an intuitive interpetation of the results above, consider the case where $K = 1$, so the possible values of the statistic are only two, namely 0 and $s_1$. In this case,

$$D^{(1)} = \alpha + \ln \frac{\Phi\left(\frac{b-s_1}{\sigma}\right) - \Phi\left(\frac{a-s_1}{\sigma}\right)}{\Phi\left(\frac{b}{\sigma}\right) - \Phi\left(\frac{a}{\sigma}\right)} - \frac{2xs_1 - s_1^2}{2\sigma^2}$$

$$= \alpha + \beta - \frac{2xs_1 - s_1^2}{2\sigma^2}$$

where

$$\beta := \ln \frac{\Phi\left(\frac{b-s_1}{\sigma}\right) - \Phi\left(\frac{a-s_1}{\sigma}\right)}{\Phi\left(\frac{b}{\sigma}\right) - \Phi\left(\frac{a}{\sigma}\right)}$$

so that we can write $x^*$ in closed form:

$$x^{*(1)} = D^{(1)-1}(0) = \frac{s_1}{2} + \frac{\sigma^2}{s_1}(\alpha + \beta)$$

which can be read as saying that the decision boundary is halfway between 0 and $s_1$, plus a term that depends on the prior bias and the effect of the boundaries of the domain of $x$ (but involves the sensitivity too, represented by $\sigma$).

Simplifying things even further, if we remove the domain boundaries (by setting $a \to -\infty$ and $b \to +\infty$), we have that $\beta \to 0$. In this case, by plugging the expression above in *Equation 6* we obtain,

$$p(\text{report noise}|s) = \Phi\left[\frac{\sigma}{s_1}\alpha - \frac{(s - s_1/2)}{\sigma}\right] \tag{7}$$

and therefore we recover a simple cumulative Normal form for the psychometric function. By looking at *Equation 7* it is clear how the prior bias $\alpha$ introduces a horizontal shift in the psychometric curve, and $\sigma$ controls the slope (but also affects the horizontal location when $\alpha \neq 0$).

## Fitting the ideal observer model to the experimental data

Independently for each rat, we infer a value of $\alpha$ and $\sigma$ by maximising the likelihood of the data under the model above. More in detail, for a given rat and a given statistic value $s$ (including 0), we call $N_s$ the number of times the rat reported 'noise', and $T_s$ the total number of trials. For a given fixed value of $\alpha$ and $\sigma$, under the ideal observer model the likelihood of $N_s$ will be given by a Binomial probability distribution for $T_s$ trials and probability of success given by the probability of reporting noise in *Equation 6*,

$$p_s(N_s|\alpha, \sigma) = \binom{T_s}{N_s} p(\text{rep. noise}|s, \alpha, \beta)^{N_s} \left(1 - p(\text{rep. noise}|s, \alpha, \beta)\right)^{T_s - N_s}$$

Assuming that the data for the different values of $s$ is conditionally independent given $\alpha$ and $\sigma$, the total log likelihood for the data of the given rat is simply the sum of the log likelihoods for the individual values of $N_s$,

$$\ln p(\{N_{s_k}\}_{k=1}^K|\alpha, \sigma) = \sum_{k=1}^K \ln p_{s_k}(N_k|\alpha, \sigma)$$

We find numerically the values of $\alpha$ and $\sigma$ that maximise this likelihood, using Matlab's mle function with initial condition $\alpha = 0.1$, $\sigma = 0.4$. Note that evaluating the likelihood for any given value of $\alpha$ and $\sigma$ requires finding $x^*$, defined as the zero of *Equation 2*. We do this numerically by using Matlab's fzero function with initial condition $x = 0$.

## Comparing the estimated sensitivity in rats to sensitivity in humans and variability in natural images

To compare quantitatively our sensitivity estimates in rat to those in humans and to the variance of the statistics in natural images reported in *Hermundstad et al., 2014*, we computed the *degree of correspondence*, as defined in *Hermundstad et al., 2014*, between these sets of numbers. Briefly, define $s_r = (1/\sigma_\beta, 1/\sigma_\theta, 1/\sigma_\alpha)$ as the array containing the rat sensitivities for the three statistics that were tested both here and by *Hermundstad et al., 2014* ($\beta = \beta_-$ and $\theta = \theta_\lrcorner$ in the notation used by *Hermundstad et al., 2014*), $s_h$ as the array containing the corresponding values for humans, and $v$ as that containing the standard deviations of the distribution of the corresponding statistics in natural images. For our comparisons, we use the values of $v$ reported by *Hermundstad et al., 2014* for the image analysis defined by the parameters $N = 2$ and $R = 32$ (i.e. the analysis used for the numbers reported in the table in *Figure 3C* in their paper). The degree of correspondence between any two of these arrays is their cosine dissimilarity:

$$c(\text{rat}, \text{human}) = \frac{s_r \cdot s_h}{\|s_r\| \cdot \|s_h\|}$$
$$c(\text{rat}, \text{images}) = \frac{s_r \cdot v}{\|s_r\| \cdot \|v\|} \quad .$$
$$c(\text{human}, \text{images}) = \frac{s_h \cdot v}{\|s_h\| \cdot \|v\|}$$

The degree of correspondence is limited by construction to values between 0 and 1, with one indicating a perfect correspondence up to a scaling factor. *Hermundstad et al., 2014* report values of 0.987–0.999 for $c(\text{human}, \text{images})$, averaging over all texture coordinates and depending on the details of the analysis.

To assess statistical significance of our values of $c$, we compare our estimated values with the null probability distribution of the cosine dissimilarity of two unit vectors sampled randomly in the positive orthant of the 3-dimensional Euclidean space. If such vectors are described, in spherical coordinates, as

$$v_1 = (\theta_1, \phi_1) \quad , \quad v_2 = (\theta_2, \phi_2)$$

with $0 \leq \theta_1, \theta_2, \phi_1, \phi_2 \leq \pi/2$, the cosine of the angle they form with each other is

$$c_{\text{null}} = v_1 \cdot v_2 = \cos(\theta_1)\cos(\theta_2) + \cos(\phi_1 - \phi_2)\sin(\theta_1)sin(\theta_2)$$

The p-values reported in the text for $c(\text{rat}, \text{human})$ and $c(\text{rat}, \text{images})$ are computed by sampling $10^7$ values of $c_{\text{null}}$, and assessing the fraction of samples with values larger than the empirical estimates.

## Acknowledgements

We acknowledge the financial support of the European Research Council Consolidator Grant project no. 616803-LEARN2SEE (DZ), the National Science Foundation grant 1734030 (VB), the National Institutes of Health grant R01NS113241 (EP) and the Computational Neuroscience Initiative of the University of Pennsylvania (VB). These funding sources had no role in the design of this study and its execution, as well as in the analyses, interpretation of the data, or decision to submit results.

## Additional information

### Funding

| Funder | Grant reference number | Author |
|---|---|---|
| FP7 Ideas: European Research Council | 616803-LEARN2SEE | Davide Zoccolan |
| National Science Foundation | 1734030 | Vijay Balasubramanian |
| National Institutes of Health | R01NS113241 | Eugenio Piasini |
| Computational Neuroscience Initiative of the University of Pennsylvania | | Vijay Balasubramanian |

The funders had no role in study design, data collection and interpretation, or the decision to submit the work for publication.

### Author contributions

Riccardo Caramellino, Conceptualization, Data curation, Formal analysis, Investigation, Methodology, Visualization, Writing - original draft; Eugenio Piasini, Conceptualization, Formal analysis, Methodology, Software, Writing – review and editing; Andrea Buccellato, Anna Carboncino, Investigation; Vijay Balasubramanian, Conceptualization, Funding acquisition, Supervision, Writing – review and editing; Davide Zoccolan, Conceptualization, Funding acquisition, Methodology, Project administration, Resources, Supervision, Writing – review and editing

### Author ORCIDs

Riccardo Caramellino ⓘ http://orcid.org/0000-0003-2201-8079
Eugenio Piasini ⓘ http://orcid.org/0000-0003-0384-7699
Vijay Balasubramanian ⓘ http://orcid.org/0000-0002-6497-3819
Davide Zoccolan ⓘ http://orcid.org/0000-0001-7221-4188

### Ethics

All animal procedures were conducted in accordance with the international and institutional standards for the care and use of animals in research and were approved by the Italian Ministry of Health and after consulting with a veterinarian (Project DGSAF 25271, submitted on December 1, 2014 and approved on September 4, 2015, approval 940/2015-PR).

### Decision letter and Author response

Decision letter https://doi.org/10.7554/eLife.72081.sa1
Author response https://doi.org/10.7554/eLife.72081.sa2

## Additional files

### Supplementary files
• Transparent reporting form

### Data availability
Experimental data are available at (Caramellino et al., 2021).

The following dataset was generated:

| Author(s) | Year | Dataset title | Dataset URL | Database and Identifier |
|---|---|---|---|---|
| Caramellino R, Piasini E, Buccellato A, Carboncino A, Balasubramanian V, Zoccolan D | 2021 | Data from "Rat sensitivity to multipoint statistics is predicted by efficient coding of natural scenes" | https://doi.org/10.5281/zenodo.4762567 | Zenodo, 10.5281/zenodo.4762567 |

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
