## [Editor Report]

This work will be of interest to neuroscientists who want to understand how visual systems are tuned to and encode natural scenes. It reports that rats share phenomenology with humans in sensitivity to spatial correlations in scenes. This shows that an earlier paper's hypothesis about efficient coding may be more broadly applicable. This work also opens up the possibility of studying this kind of visual tuning in an animal where invasive techniques can be used to study this neural origins of this sensitivity and its development.

---

## [Decision Letter]

**Decision letter after peer review:**

Thank you for submitting your article "Rat sensitivity to multipoint statistics is predicted by efficient coding of natural scenes" for consideration by *eLife*. Your article has been reviewed by 2 peer reviewers, and the evaluation has been overseen by a Reviewing Editor and Timothy Behrens as the Senior Editor. The reviewers have opted to remain anonymous.

Essential revisions:

All reviewers thought that the paper was exciting, but needed some revision to clarify the results and presentation.

1) It would be useful if the manuscript scaled back claims about the alignment between the human and rat data slightly to make them more comparable with the results presented in the paper, and also to emphasize the ranking of sensitivities, qualitatively, is what's established, rather than a precise quantitative match to the full correlation matrix. Some specific points along these lines are:

– It is unclear why the ranking in rat sensitivity is evidence for efficient coding. In Hermundstad et al., 2014, efficient coding was established by comparing the image-based precision matrix with the human perceptual isodiscrimination contours. There is no such comparison here.

– Revision prompt (1a) Claims should be softened slightly.

– The previous paper emphasized that the difference of perceptual sensitivity between horizontal/vertical edges and diagonal edges is not merely an "oblique effect": Horizontal and vertical pairwise correlation share an edge, while pixels involved in diagonal pairwise correlations only share a corner. One wonders whether rats show any sensitivity difference between horizontal/vertical edges and diagonal edges. The manuscript in its current form misses this important comparison. Without showing this, the rat sensitivity does not fully reproduce the trend previously observed in humans. It seems like acquiring new data from the rats is prohibitively time-consuming, so again, the claims of the paper should be softened a bit.

Revision prompt (1b) – If possible, it would be useful to see a comparison of the rat sensitivity to different 2-point correlations, and a note about whether it matches the human data or not.

Revision prompt (1c) – It would be very helpful if the authors can generate analysis as in Figure 3B or 3C in Hermundstad et al., 2014 (3C is maybe easier?). If such analysis is possible, then it shows that the rat sensitivity also quantitatively matches the results from efficient coding. Again, if this is prohibitive, claims should be softened.

2) One part of the analysis was unclear: Why does it work with this theory to find the sensitivity only to positive parity values?

It seemed surprising that one would not need to test negative pairwise correlations, negative 3-point correlations, or negative 4-point patterns. For the 3-point glider, in particular, the large correlated patches (triangles) change contrast when parity is inverted, so it is the only correlational stimulus in this set that inverts contrast under parity inversion (besides the trivial 1-point glider). Given the light-dark asymmetries of the natural world, it would seem possible that the three-point sensitivity depends (strongly?) on the parity. This seems to be true of some older point-statistic discrimination tasks in humans (from Chubb?), where the number of black pixels (rather than merely dark gray) seemed to account for human discrimination thresholds. The parity of 3-point gliders clearly makes an impact on motion perception when these are looked at in space-time (i.e., Hu and Victor and various subsequent work in flies and fish), and the percept strength is also different for positive vs. negative parity. So, given the contrast inversion asymmetry in 3-point gliders and prior work on light-dark asymmetries in discriminability, it seems one needs to test whether sensitivity is the same under positive and negative parity for these types of spatial correlations. If the authors contend that this is not necessary given the efficient coding hypothesis being tested, some discussion is warranted of light-dark asymmetries in natural scenes and in this suite of stimuli, and why they are neglected in this framework (if that's the case).

3) Figure 3 needs revision for clarity. All reviewers found the layout confusing. Perhaps the authors could find a clearer way to present the results, using more figure panels.

4) The luminance values listed for the visual stimuli seem rather odd, since the mean luminance is not the average of the max and min luminance (the light and dark pixels). This seems to imply that these patterns do contain not equal numbers of light and dark pixels, which they should for all the 2, 3, and 4 point glider stimuli. It's not clear how this is consistent with the described experiments. Please clarify this point in the text.

---

## [Author Response]

Essential revisions:All reviewers thought that the paper was exciting, but needed some revision to clarify the results and presentation.1) It would be useful if the manuscript scaled back claims about the alignment between the human and rat data slightly to make them more comparable with the results presented in the paper, and also to emphasize the ranking of sensitivities, qualitatively, is what's established, rather than a precise quantitative match to the full correlation matrix. Some specific points along these lines are:– It is unclear why the ranking in rat sensitivity is evidence for efficient coding. In Hermundstad et al., 2014, efficient coding was established by comparing the image-based precision matrix with the human perceptual isodiscrimination contours. There is no such comparison here.– Revision prompt (1a) Claims should be softened slightly.

We thank the reviewers for underscoring the difference between, on the one hand, a quantitative comparison of the sensitivity to the variance of the statistics in natural images, and on the other hand a more qualitative comparison of their rank ordering. In our initial submission, we built our argument based on the rankings in order to better connect not only with Hermundstad et al., 2014, but also with earlier human psychophysics results on the same task (Victor and Conte 2012), where there was no comparison with natural image statistics and therefore only the qualitative ranking among sensitivities was examined. We also note that Hermundstad et al., do, in fact, make ample use of the rank-ordering agreement between natural image statistics and human sensitivity in order to support their argument (“rank-order”, or similar locutions, are used six times between the results and the discussion). In this sense, while it is true that “In Hermundstad et al., 2014, efficient coding was established by comparing the image-based precision matrix with the human perceptual isodiscrimination contours”, it is also true that the rank ordering was presented as part of the evidence for efficient coding.

Having said this, we nevertheless agree that our argument can be strengthened by presenting both approaches, qualitative and quantitative. We have now added a new figure (Figure 3), where we compare our estimates of psychophysical sensitivity in rats with the corresponding values for human psychophysics and natural image statistics reported in Hermunstad 2014 (note that we were only able to compare three out of the four statistics that we tested, as Hermunstad et al., did not consider 1-point correlations). The comparisons in Figure 3 (and the related quantitative measures reported in the text, lines 146-160) reveal a strong quantitative match, similar to that between the human psychophysics and the image statistic data.

Finally, in response to a specific point raised by Reviewer 2, point 1 (“Hermundstad et al., 2014 did not include 1^st^-order at all.”), the 1^st^ order statistic γ was indeed only studied in Victor and Conte 2012, which only contained phychophysics data, and not in Hermundstad et al., 2014, which connected psychophysics with natural image statistics. Indeed, it is not possible to analyze the variability of γ in natural images with the method established by Hermundstad et al., 2014, because each image is binarized in such a way to guarantee that γ=0 by construction. In this sense, like the use of qualitative ranking discussed above, γ was included to better reflect the approach in Victor and Conte. Moreover, we wanted to include a sensory stimulus condition that we were sure the animals could detect well, in order to ensure that any failure to learn or perform the task was due to limitations in sensory processing and not in the learning or decision-making process. Before performing our experiments, the only statistic that we were confident the rats could be trained to distinguish from noise was γ [Tafazoli et al., 2017, Vascon et al., 2019], and therefore it made sense to include it in the experimental design. We have modified the Results (lines 90-93, 104-108), the Methods (316-321) and the Discussion (212-218) to express this point more clearly.

– The previous paper emphasized that the difference of perceptual sensitivity between horizontal/vertical edges and diagonal edges is not merely an "oblique effect": Horizontal and vertical pairwise correlation share an edge, while pixels involved in diagonal pairwise correlations only share a corner. One wonders whether rats show any sensitivity difference between horizontal/vertical edges and diagonal edges. The manuscript in its current form misses this important comparison. Without showing this, the rat sensitivity does not fully reproduce the trend previously observed in humans. It seems like acquiring new data from the rats is prohibitively time-consuming, so again, the claims of the paper should be softened a bit.Revision prompt (1b) – If possible, it would be useful to see a comparison of the rat sensitivity to different 2-point correlations, and a note about whether it matches the human data or not.

When designing our experiment, we prioritized collecting data for the other statistics as they were closer to the extremes of the measured sensitivity values, therefore offering a clearer signal for a comparison with rat data. For instance, had we found better sensitivity to 3- or 4-point statistics than to (horizontal) 2-point statistics, this would have been a very clear sign that perceptual sensitivity in rat is organized differently than in humans. Conversely, we reasoned that a comparison based on 2-point diagonal instead of 2-point horizontal would have been more easily muddled and made inconclusive by the experimental noise that we expected to observe in rats. We agree that, given the high precision of the quantitative match between rats, humans and image statistics now highlighted by the new Figure 3, it would be interesting to test rats also for their sensitivity to diagonal 2-point correlations and check whether they matched the pattern exhibited by humans. However, as the editor rightly surmises, acquiring new data at this stage would indeed be exceedingly time consuming. Therefore, we have modified the text to better highlight that we did not seek to replicate this particular result in Hermundstad et al., 2014 (as well as that we could not test as many correlation patterns as in Hermundstad et al., 2014 more generally, due to practical and ethical constraints). We also note that, since we did not test 2-point diagonal, we can’t draw conclusions similar to those in Hermundstad 2014 about the difference of an effect due to efficient coding and one due to a hypothetical oblique effect for the specific 2-point horizontal vs. diagonal comparison. These points are now all brought up in the Discussion of our revised manuscript (lines 189-208). It is also worth noting that the oblique effect was a minor point of the Hermundstad et al., paper and the main arguments did not hinge on it.

Revision prompt (1c) – It would be very helpful if the authors can generate analysis as in Figure 3B or 3C in Hermundstad et al., 2014 (3C is maybe easier?). If such analysis is possible, then it shows that the rat sensitivity also quantitatively matches the results from efficient coding. Again, if this is prohibitive, claims should be softened.

Thank you for the suggestion. As mentioned above, we have now added a new figure (Figure 3) where we compare rat sensitivity, human sensitivity, and image statistic data in a way similar to Figure 3B in Hermundstad 2014, for the image statistics that were tested in both our experiment and in Hermundstad et al., 2014. We have also computed the “degree of correspondence” *c* between rat and image data and between rat and human data, using the definition of this metric introduced by Hermundstad et al., and reported by them in Figure 3C and in the main text. The degree of correspondence captures quantitatively the excellent match between rat, human and image data, with c(rat, image)=0.986 and c(rat, human)=0.990, where 0≤c≤1, and c=1 indicates perfect match. These results are reported in the Results section of the updated manuscript (lines 146-160).

2) One part of the analysis was unclear: Why does it work with this theory to find the sensitivity only to positive parity values?It seemed surprising that one would not need to test negative pairwise correlations, negative 3-point correlations, or negative 4-point patterns. For the 3-point glider, in particular, the large correlated patches (triangles) change contrast when parity is inverted, so it is the only correlational stimulus in this set that inverts contrast under parity inversion (besides the trivial 1-point glider). Given the light-dark asymmetries of the natural world, it would seem possible that the three-point sensitivity depends (strongly?) on the parity. This seems to be true of some older point-statistic discrimination tasks in humans (from Chubb?), where the number of black pixels (rather than merely dark gray) seemed to account for human discrimination thresholds. The parity of 3-point gliders clearly makes an impact on motion perception when these are looked at in space-time (i.e., Hu and Victor and various subsequent work in flies and fish), and the percept strength is also different for positive vs. negative parity. So, given the contrast inversion asymmetry in 3-point gliders and prior work on light-dark asymmetries in discriminability, it seems one needs to test whether sensitivity is the same under positive and negative parity for these types of spatial correlations. If the authors contend that this is not necessary given the efficient coding hypothesis being tested, some discussion is warranted of light-dark asymmetries in natural scenes and in this suite of stimuli, and why they are neglected in this framework (if that’s the case).

Before addressing the reviewer’s point, we should first clarify that the range of values of the 3-point statistic used in our experiment in fact spans the negative, rather than positive, half of the space of possibilities in the parameterization of Victor and Conte, 2012. We reported these as positive values in our initial submission due to an inversion of the 3-point axis in our code. This is simply a matter of convention and does not affect any of the arguments made in the paper, or the reviewer’s point, but we wanted to clarify this first. We have now explained in the Methods section (lines 334-338) that, although we still refer to 3-point intensities using positive numbers, if the reader is interested in connecting formally to the system of coordinates in Victor and Conte 2012, the sign of the values of 3-point statistics we report should be inverted.

In humans, the answer to the question raised by the reviewer is already known: Victor and Conte report that “consistently across subjects, thresholds for negative and positive variations of each statistic are closely matched” (Victor and Conte 2012, caption to Figure 7). Similarly, Hermundstad et al., 2014 remark on the same phenomenon and investigate it specifically (Hermundstad et al., 2014, Figure 3 —figure sup)

Moreover, even foregoing the above argument about human equal sensitivity to the positive and negative variations in the statistics, we observe the following with respect to the mention of the contrast inversion asymmetry in 3-point gliders, in relation to the light-dark asymmetry in natural scenes. Dominance of OFF responses (elicited by dark spots on a light background) has been reported in mammals, including primates, cats, and, more recently, rodents (Liu and Yao 2014; Xing, Yeh, and Shapley 2010; Yeh, Xing, and Shapley 2009; Williams et al., 2021). Therefore, if rats unlike humans had different sensitivity to positive and negative 3-point statistics, one would expect that the sensitivity to the negative 3-point correlations would be the highest of the two (as negative intensities corresponds to dark triangular patterns on white background). Since we are interested in the hypothesis that the 3-point configuration is the statistic with the lowest sensitivity of those we tested, by testing negative intensities we are choosing the stricter test, whereas testing positive values would risk biasing the experiment towards the desired conclusion. Indeed, this was the reason why the negative half of the 3-point axis was chosen in the first place.

As for the reason we used positive intensity values of the 2-point and 4-point statistics, this choice was dictated by the need of testing rats with textures containing features that were large enough to be processed by their low-resolution visual system. In fact, rat visual acuity is much lower than human acuity and, while positive 2-point and 4-point correlations give rise to, respectively, thick oriented stripes and wide rectangular blocks made of multiple pixels with the same color, negative intensities produce higher spatial frequency patterns, where color may change every other pixel see Figure 2A in Hermundstad et al., 2014. Therefore, using negative 2-point and 4-point statistics would have introduced a possible confound, since low sensitivity to these textures could have been simply due to the low spatial resolution of rat vision. Finally, in the case of the 1-point statistic, positive intensity values were chosen because they yield patterns that are brighter than white noise and, as such (we reasoned), would be highly distinguishable from white noise, given the high sensitivity of rat V1 neurons to increases of luminance.

All these explanations are now provided in the Methods section (lines 322-338) and a thorough discussion of the possible impact of our stimulus choices (both at the level of texture type and polarity) on our conclusions is now presented in the Discussion of our revised manuscript, including the rationale behind testing only on either positive or negative values of each given statistic (lines 189-249).

3) Figure 3 needs revision for clarity. All reviewers found the layout confusing. Perhaps the authors could find a clearer way to present the results, using more figure panels.

Thank you for this valuable suggestion. This figure (that in our revised manuscript has become Figure 4) has now been redesigned from scratch for better clarity. We still kept all data in a single panel in order to enable easy comparisons between conditions with same test statistic but different training, or same training and different test statistics.

4) The luminance values listed for the visual stimuli seem rather odd, since the mean luminance is not the average of the max and min luminance (the light and dark pixels). This seems to imply that these patterns do contain not equal numbers of light and dark pixels, which they should for all the 2, 3, and 4 point glider stimuli. It's not clear how this is consistent with the described experiments. Please clarify this point in the text.

Thank you for noticing this inconsistency with the luminance values reported in our Method section. In fact, while the minimal and maximal luminance values of the display were correctly reported in our sentence, the luminance that we reported as corresponding to mid-gray was incorrect. The actual value is, as it should, halfway between the maximum and minimum (average among monitors is equal to 61±8 cd/mm). This error was due to the fact that we erroneously reported the luminance of pixel intensity level 128 without taking into account the linearization of the pixel/luminance curve that we carried out before presenting the stimuli. We have now corrected this error in our revised Methods (lines 355-357), where we simply report the average maximal and minimal luminosity levels of the monitors.